# Cytomegalovirus DNA Loads in Organs of Congenitally Infected Fetus

**DOI:** 10.3390/v16060891

**Published:** 2024-05-31

**Authors:** Kuniaki Toriyabe, Asa Kitamura, Makoto Ikejiri, Ryotaro Hashizume, Maki Nakamura, Emi Teramoto, Hiroki Takeuchi, Eiji Kondo, Tomoaki Ikeda

**Affiliations:** 1Department of Obstetrics and Gynecology, Mie University Graduate School of Medicine, Tsu 514-8507, Mie, Japan; 2Department of Obstetrics and Gynecology, National Hospital Organization Mie Chuo Medical Center, Tsu 514-1101, Mie, Japan; 3Department of Clinical Laboratory, Mie University Hospital, Tsu 514-8507, Mie, Japan; 4Department of Pathology and Matrix Biology, Mie University Graduate School of Medicine, Tsu 514-8507, Mie, Japan; 5Department of Genomic Medicine, Mie University Hospital, Tsu 514-8507, Mie, Japan

**Keywords:** fetal autopsy, congenital cytomegalovirus, cytomegalic inclusion body, cytomegalovirus DNA

## Abstract

Congenital cytomegalovirus (cCMV) infection poses significant risks to fetal development, particularly affecting the nervous system. This study reports a fetal autopsy case, examining cCMV infection and focusing on CMV DNA measurements in various fetal organs before formalin fixation, a novel approach for comprehensive CMV DNA evaluations in fetal organs affected by cCMV. A 20-week-old male fetus was diagnosed with cCMV following the detection of CMV DNA in ascites obtained via abdominocentesis in utero. After the termination of pregnancy, multiple organs of the fetus, including the cerebrum, thyroid gland, heart, lungs, liver, spleen, kidneys, and adrenal glands, were extracted and examined for CMV DNA loads using a real-time polymerase chain reaction. Histopathological examination involved hematoxylin–eosin and CMV-specific immunostaining. A correlation was found between CMV DNA loads and pathology, with higher CMV-infected cell numbers observed in organs positively identified with both staining methods, exhibiting CMV DNA levels of ≥1.0 × 10^4^ copies/mL, compared to those detected solely by CMV-specific immunostaining, where CMV DNA levels ranged from 1.0 × 10^3^ to 1.0 × 10^4^ copies/mL. These results highlight a quantifiable relationship between the organ infection extent and CMV DNA concentration, providing insights into cCMV pathogenesis and potentially informing future diagnostic and therapeutic strategies for cCMV infection.

## 1. Introduction

Cytomegalovirus (CMV) is a pathogen associated with mother-to-child transmission that is known to cause damage to the fetal nervous system. Fetal congenital cytomegalovirus (cCMV) infection occurs due to transplacental CMV transmission. According to a large study in Japan, the CMV antibody prevalence among pregnant women was 64.2% [1]. Alternatively, another large study in Japan reported that the incidence rate of neonatal cCMV infection was 0.3% [2]. CMV can infect various organs, including the brain, eyes, inner ears, salivary glands, thyroid gland, thymus, heart, lungs, stomach, intestine, liver, spleen, pancreas, kidneys, adrenal glands, epididymis, and testis [3,4,5,6]. The kidneys, lungs, liver, and salivary glands are the most commonly affected organs above [7]. In each organ, CMV-infected cells are identified based on the presence of cytomegalic inclusion bodies (CIBs). There are two types of CIBs: intranuclear amphophilic and cytoplasmic basophilic granular inclusion bodies. The former is a better known diagnostic tool for CMV infections. Additional pathological tools are sometimes used, as well as conventional hematoxylin–eosin (HE) stains, such as CMV-specific immunostaining and CMV DNA in situ hybridization, sometimes being implemented to detect CIBs in infected cells [4,7].

In the current study, we measured CMV DNA in the organs of a fetus with congenital CMV infection using a real-time polymerase chain reaction (PCR). To our knowledge, there are no reports on CMV DNA measurements in multiple fetal organs with cCMV infection. In this fetal autopsy case study of cCMV infection, we performed CMV DNA measurements in fetal organ samples before formalin fixation and a pathological examination of fetal organs after formalin fixation.

## 2. Materials and Methods

This study focused on a male fetus at 20 weeks of gestational age (GA) who developed ascites at 16 weeks of GA and underwent abdominocentesis through the maternal abdominal wall at 18 weeks of GA. CMV DNA was identified in the ascites sample (5.9 × 10^5^ copies/mL), leading to a diagnosis of cCMV infection. The volume of amniotic fluid was normal (the maximum vertical pocket of amniotic fluid, 3.6 cm). There was no placental edema (the thickness of placenta, 3.7 cm). The fetus showed neither growth restriction (estimated body weight, 0 standard deviation) nor anemia (the peak systolic velocity of middle cerebral artery, 1 multiple of median). His mother was 26 years old and was para 1. Her anti-CMV antibodies were strongly positive for immunoglobulin (Ig) G and weakly positive for IgM, and IgG antibody avidity was high. Viral strains of glycoprotein H subunits in amniotic fluid and maternal serum samples were both the AD type. Following the decision to terminate the pregnancy, the mother underwent a vaginal abortion (in Japan, artificial abortions for maternal health protection at less than 22 weeks of GA are permissible by law). The fetus weighed 476 g and measured 25 cm in length. Both the head and abdominal circumferences were recorded at 18 cm each. Notably, the abdomen was distended, and ascitic fluid totaling 43 mL was extracted.

Purpura was observed throughout the body (Figure 1). The placental weight was 291 g, and the length of the umbilical cord was 23 cm. An anti-CMV antibody in the umbilical cord blood was positive for IgM and CMV DNA was identified in the umbilical cord blood (9.9 × 10^4^ copies/mL). His parents provided informed consent for the extraction and testing of his organs. Two and a half hours after delivery, we extracted the cerebrum, thyroid gland, thymus, heart, lungs, stomach, intestine, liver, spleen, pancreas, kidneys, and adrenal gland.

We collected 0.1–0.2 g of samples from the organs (excluding the thymus, stomach, intestine, and pancreas) and soaked them in formalin. CMV DNA was extracted from samples using a QIAamp DNA Mini Kit (Qiagen, Hilden, Germany). Real-time PCR was performed to measure CMV DNA load as previously described [8]. After formalin fixation of the extracted organs, pathological examinations were performed using HE and CMV-specific immunostaining. We took five sections from the cerebrum, three from the thyroid gland, one from the thymus, four from the heart, six (three each side) from the lungs, two from the stomach, three from the intestine, one from the liver, four from the spleen, two from the pancreas, four (two each side) from the kidneys, and seven (three from left side and four from right side) from the adrenal glands. We examined the full extent of all organ sections at a 400 high-power field (HPF). We classified an organ where 0–3 CMV-infected cells were found at 400 HPF as level 1. Alternatively, we classified an organ where 0–6 infected cells were found as level 2. We used a Dako anti-CMV monoclonal antibody (clone CCH2 + DDG9) targeting an immediate early and early antigen (Agilent, Santa Clara, CA, USA).

## 3. Results

CMV DNA was detected in all tested organ samples. CMV DNA loads in each organ sample are presented in Table 1. CMV DNA of ≥1.0 × 10^3^ copies/µgDNA was identified in all organ samples. In the samples from the thyroid gland, lungs, liver, and kidneys, CMV DNA levels were found to be ≥1.0 × 10^4^ copies/µg DNA.

The pathological results for each organ are presented in Table 1. CMV-infected cells showed both HE staining and CMV-specific immunostaining in the thyroid gland (level 2), lungs (level 1), liver (level 2), and kidneys (level 2) (Figure 2). CMV-infected cells were detected only by CMV-specific immunostaining in the cerebrum, thymus, heart, spleen, pancreas, and adrenal glands (all level 1) (Figure 3 and Figure 4). In the cerebrum, no strong pathological findings of tissue necrosis, microglial nodules, or other inflammatory changes were observed. CMV-infected cells were not found in the stomach or intestine with HE staining or CMV-specific immunostaining.

In the tested samples of the organs where CMV-infected cells were found both with HE stains and with CMV-specific immunostaining, CMV DNA of ≥1.0 × 10^4^ copies/µg DNA was detected. Alternatively, in the other tested samples of the organs where CMV-infected cells were found only with CMV-specific immunostaining, CMV DNA of 1.0 × 10^3^–1.0 × 10^4^ copies/µg DNA was detected.

## 4. Discussion

In the current fetal autopsy case study, a correlation between pathology and CMV DNA loads in the organs of fetuses with cCMV infection was suggested. A report on CMV DNA measurements in multiple organs of patients with human immunodeficiency virus (139 organs in 11 patients) has been published [9]. The authors compared CMV DNA loads in organs where CMV-infected cells were identified through HE staining or CMV-specific immunostaining to those organs without such findings. They reported a strong correlation between the presence of CMV-infected cells and the detection of CMV DNA via quantitative competitive PCR in the evaluated organs. Notably, higher CMV DNA loads were observed in organs where CMV-infected cells were present compared to organs where infected cells were absent. This was attributed to the significantly higher number of CMV-infected cells in the organs where the infected cells were detected.

Although we could not compare CMV DNA loads between the organs where CMV-infected cells were found (cerebrum, thyroid gland, heart, lungs, liver, spleen, kidneys, and adrenal glands) and other organs where infected cells were not found (stomach and intestine), the same idea could be used to interpret our findings.

In the present case study, compared to the fetal organs where CMV-infected cells were found only with CMV-specific immunostaining, the other fetal organs where CMV-infected cells were found, both with HE staining and CMV-specific immunostaining, had 10 times more CMV DNA loads. The higher CMV DNA loads might be due to the higher number of CMV-infected cells in the organs of fetuses with cCMV infection. It was hypothesized that the quantity of CMV-infected cells is greater in fetal organs identified by both HE staining and CMV-specific immunostaining, in comparison to organs where CMV-infected cells were detected solely via CMV-specific immunostaining. The subsequent objective involved assessing whether CMV DNA loads in organs identified exclusively through CMV-specific immunostaining, or in those organs where infected cells were not discernible with either HE stains or CMV-specific immunostaining, are elevated.

The prevalence of CMV shedding in any body fluid of CMV-seropositive pregnant women was reported to be in the range 0–42.5% [10]. CMV shedding could aid in identifying CMV-seropositive pregnant women at an increased risk of having a fetus with cCMV infection. In the present case, although his mother was considered as having non-primary infection by her serology, we did not go so far as to test her body fluids for CMV shedding. Since the neuropathogenesis of brain injury related to cCMV infection is poorly understood, the characteristics and pathogenetic mechanisms of encephalic damage in cCMV infection are being studied. A previous study suggested a preferential CMV tropism for both neural stem/progenitor cells and neuronal committed cells [11]. The current study focused only on the presence of CMV-infected cells in the cerebrum sample and did not focus on distribution; since CMV distribution is important in cCMV-induced brain damage, it is worth re-examining it in the cerebrum block. That is our next task for the future.

A limitation of this case study is the inability to examine the correlation between pathology and CMV DNA load across all extracted fetal organs due to the absence of samples from the thymus, stomach, intestine, and pancreas. Future pathological autopsy cases focusing on cCMV infection should include the collection of samples from these organs to fully assess the correlation between pathology and CMV DNA loads in all extracted fetal organs. Another important limitation is that we did not search for pathology in the auditory pathway. Although cCMV infection is well known to involve the auditory pathway and cause hearing impairment, fetal cases are almost the only pathological candidate for a clinical specimen [12]. Future pathological autopsy cCMV cases should include the collection of samples from the auditory pathway to assess the pathology of auditory impairment.

In conclusion, we found a potential correlation between pathology and CMV DNA loads in the organs of a fetus with cCMV infection. Exploring this avenue in the future could deepen our understanding of the pathogenesis of cCMV infections and facilitate the development of innovative treatment strategies.

## Figures and Tables

**Figure 1 viruses-16-00891-f001:**
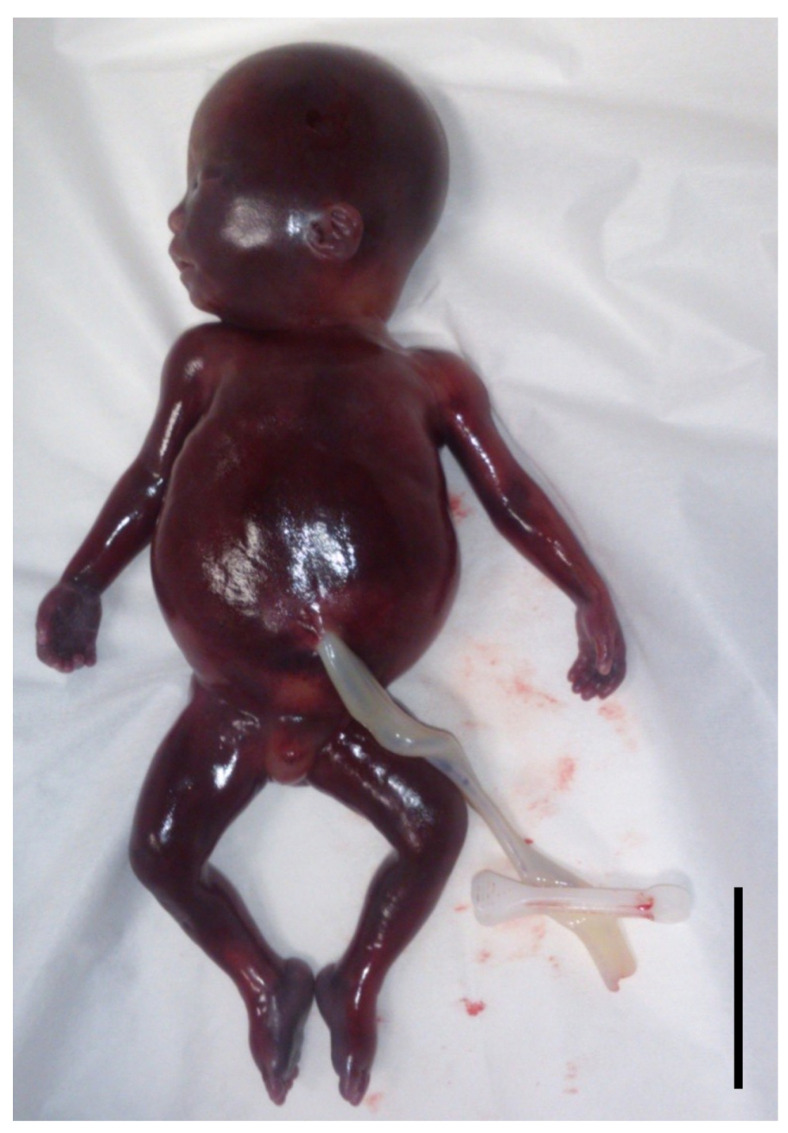
Appearance of the case before fetal autopsy. His abdomen was distended by ascites, and purpura was observed all over the body (actual size, scale bar: 5 cm).

**Figure 2 viruses-16-00891-f002:**
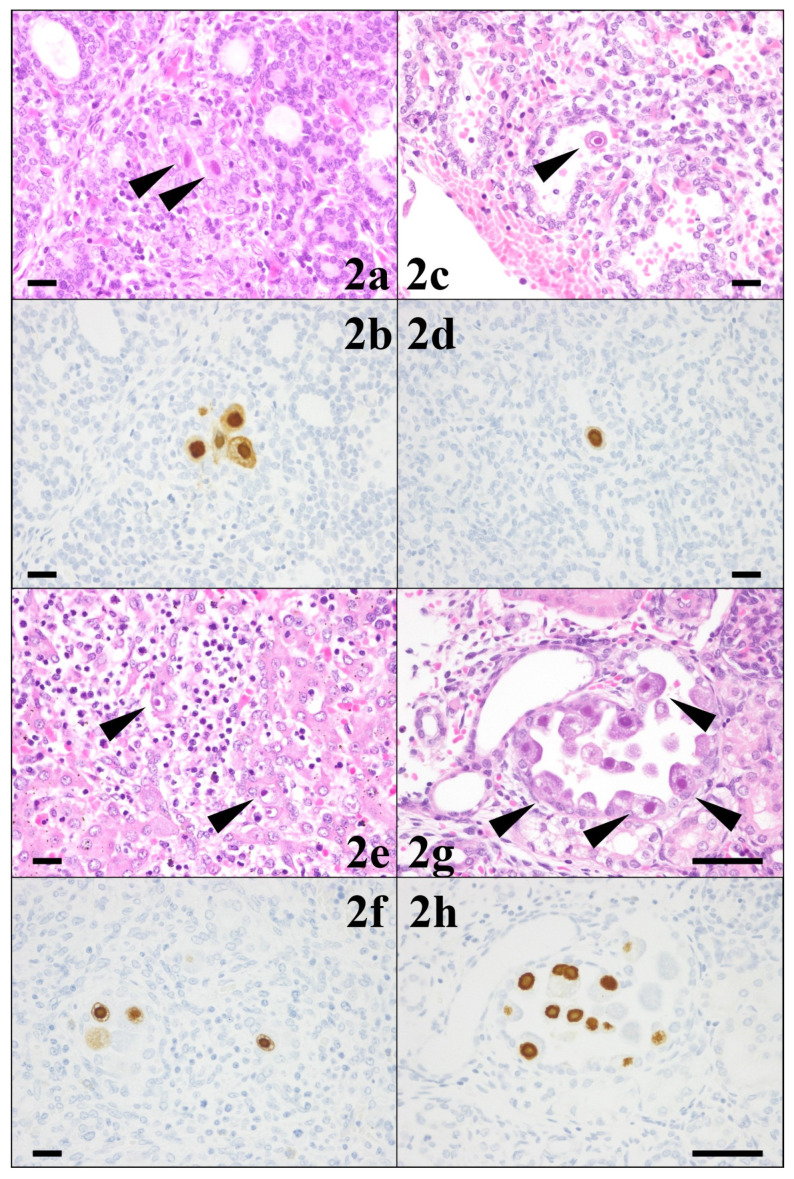
Pathology of the thyroid gland (**2a**,**2b**), lung (**2c**,**2d**), liver (**2e**,**2f**), and kidney (**2g**,**2h**) ((**2a**–**2f**), 40 HPF; (**2g**,**2h**), 100 HPF, scale bar: 50 µm). In all four fetal organs, cytomegalovirus (CMV)-infected cells with cytomegalic inclusion bodies (arrowhead) were found both with hematoxylin–eosin stains (upper row) and with CMV-specific immunostaining (Dako anti-CMV monoclonal antibody) (lower row).

**Figure 3 viruses-16-00891-f003:**
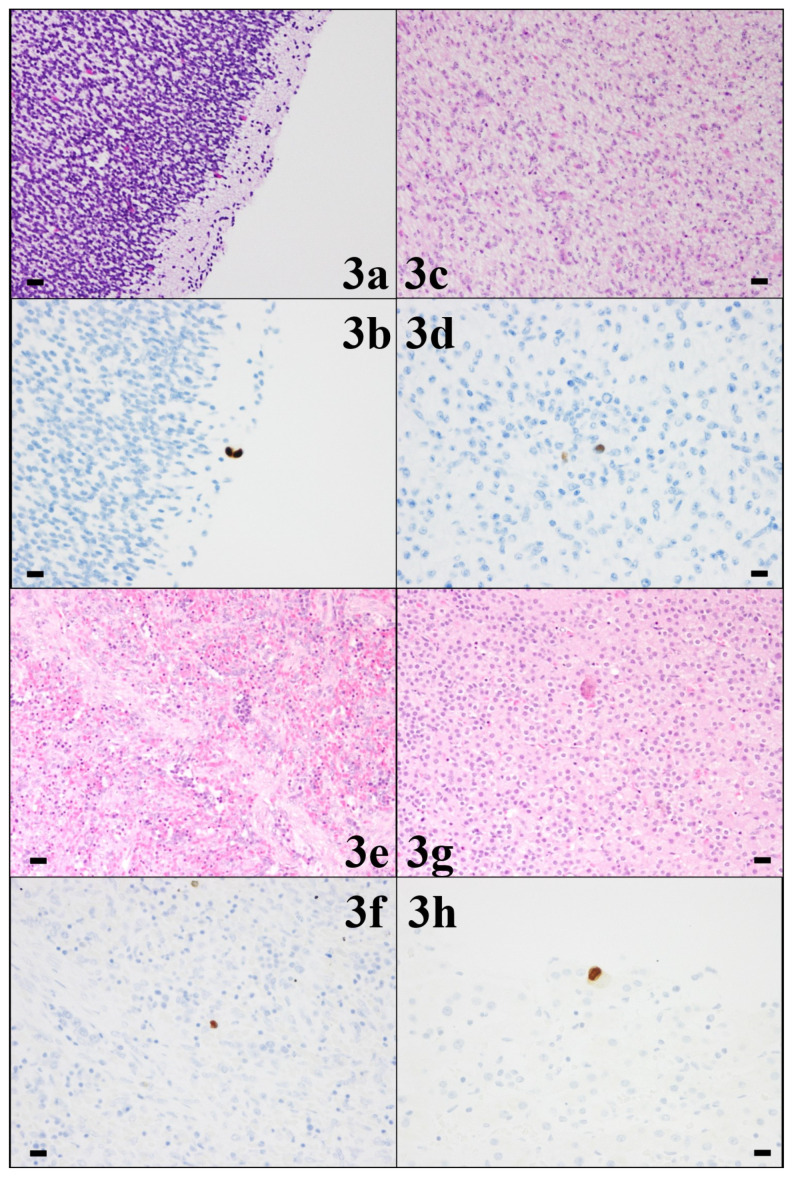
Pathology of cerebrum (**3a**,**3b**), heart (**3c**,**3d**), spleen (**3e**,**3f**), and adrenal gland (**3g**,**3h**) (10 HPF, scale bar: 50 µm). In all four fetal organs, cytomegalovirus (CMV)-infected cells with cytomegalic inclusion bodies were found only with CMV-specific immunostaining (Dako anti-CMV monoclonal antibody) (lower row).

**Figure 4 viruses-16-00891-f004:**
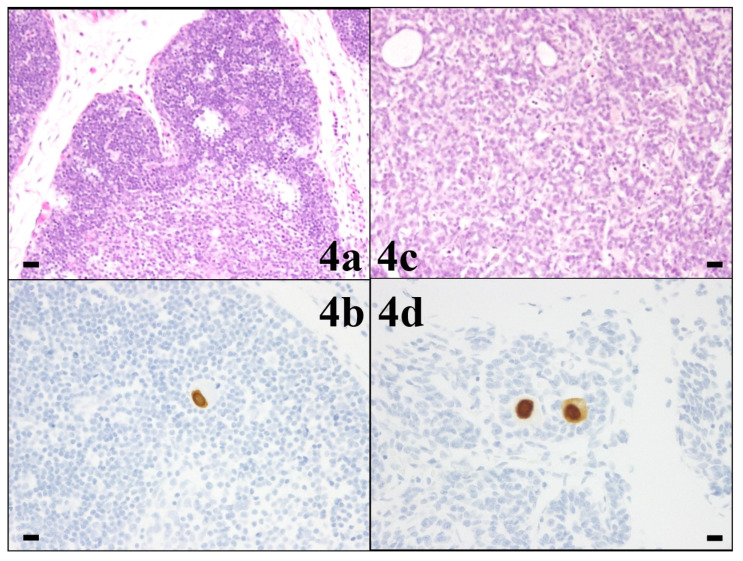
Pathology of thymus (**4a**,**4b**) and pancreas (**4c**,**4d**) (10 HPF, scale bar: 50 µm). In both organs, cytomegalovirus (CMV)-infected cells with cytomegalic inclusion bodies were found only with CMV-specific immunostaining (Dako anti-CMV monoclonal antibody) (lower row).

**Table 1 viruses-16-00891-t001:** Presence of cytomegalovirus (CMV)-infected cells with hematoxylin–eosin stains and with CMV-specific immunostaining and loads of CMV DNA in each fetal organ sample.

Fetal Organ Sample (n *)	CMV-Infected Cells with HE Stains (Level **)	CMV-Infected Cells with CMV-Specific Immunostaining (Level **)	CMV DNA Loads (Copies/µg DNA)
Thyroid gland (3)	+ (2)	+ (2)	5.7 × 10^4^
Lung (6)	+ (1)	+ (1)	5.5 × 10^4^
Liver (1)	+ (2)	+ (2)	4.9 × 10^4^
Kidney (4)	+ (2)	+ (2)	5.7 × 10^4^
Cerebrum (5)	-	+ (1)	1.5 × 10^3^
Heart (4)	-	+ (1)	1.1 × 10^3^
Spleen (4)	-	+ (1)	1.5 × 10^3^
Adrenal gland (7)	-	+ (1)	2.4 × 10^3^
Thymus (1)	-	+ (1)	NE
Pancreas (2)	-	+ (1)	NE
Stomach (2)	-	-	NE
Intestine (3)	-	-	NE

Cytomegalovirus, CMV; Hematoxylin–eosin, HE; Not examined, NE. * Number of organ sections examined. ** In level 1 organ, we found 0–3 CMV-infected cells and in level 2 organ, we found 0–6 infected cells at 400 HPF. In “+” organ, CMV-infected cells were found and in “-” organ, CMV-infected cells were not found.

## Data Availability

The datasets used and/or analyzed in this case study are available from the corresponding author upon reasonable request.

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
