# Peer review of "Cytomegalovirus DNA Loads in Organs of Congenitally Infected Fetus"

_viruses, 2024, doi:10.3390/v16060891_

Round 1

Reviewer 1 Report

Comments and Suggestions for Authors

Toriyabe et al. report a detailed evaluation of a case of congenital cytomegalovirus (cCMV) in a 20 week-gestation male fetus.

The manuscripot should better indicate briefly whether disease was diagnosed above and beyond CMV infection, as well as whether any clinical signs or symptoms were assessed by ultrasound, MRI or other methods.  The Introduction indicates that the fetus had a distended abdomen and exhibited purpura, which is shown in figure 1 as extreme, and the presence of virus and viral immunopathology in many internal organs is consistent with most severe cCMV disease. The observation that all of organs tested were positive for CMV DNA leaves a possibility that this is because severe cCMV infection includes viremia that was not directly evaluated in the fetal circulation or in the cord blood.  

First, it would be beneficial to include any clinical assessment of the actual disease level that would be expected had this pregnancy gone to term, particularly with the notable CMV-associated immunopathology in the cerebrum. Second, severe cCMV disease is systemic, as this appears, but the field has had limited opportunity to evaluate neuronal infection that predominates in more common cCMV disease settings.  It seems an additional oversight that more effort was not expended to examine neurons by immunohistochemistry in various locations including those involved in auditory perception and vision. Thus, oversights in the depth of evaluation here reduces the scholarship and overall impact of the work.

Nevertheless, authors should be encouraged to place this case in context by reflecting further on the case evaluations that have been presented in the past. See: doi: 10.1007/s10571-022-01258-9, doi: 10.1002/rmv.2399 and doi: 10.3390/ijms25052636 and references within to prior case studies.

Comments on the Quality of English Language

none additional

Reviewer 2 Report

Comments and Suggestions for Authors

abstract / introduction: CMV and cCMV epidemiology in Japan? Rationale for abortion, local criteria?

M&M: informed consent for publication of fetus by parents provided? anti-CMV antibody target? How were histology samples analyzed for infected cells? How many samples were analyzed etc.? 

Fig 1.: Please consider if e.g.  eyes should be covered? Scale bar is missing. Size of image nescessary?

Fig 2-4. low image resolution? Please indicate/highlight cells with inclusion bodies. The reader cannot extrapolate the positioning of infected cells within the organs. Are the cells close to blood vessels? Which cells are nearby? At least a combination of CD45 and CMV staining would be needed to provide some insight into inflammatory response. Scale bar missing, labels of figure items to small. Figure legend provide "result" instead of describition what has been done. 

Table 1: How many slides / viewfield were analysed and how many cells were detected in each sample? Data was not quantified.

Comments on the Quality of English Language

editing of English required

Round 2

Reviewer 1 Report

Comments and Suggestions for Authors

Authors have addressed reviewers' concerns fully.